# Rescue of a Rotenone Model of Parkinson’s Disease in *C. elegans* by the Mitochondrial Na^+^/Ca^2+^ Exchanger Inhibitor CGP37157

**DOI:** 10.3390/ijms26073371

**Published:** 2025-04-04

**Authors:** Silvia Romero-Sanz, Elena Caldero-Escudero, Pilar Álvarez-Illera, Jaime Santo-Domingo, Sergio de la Fuente, Paloma García-Casas, Rosalba I. Fonteriz, Mayte Montero, Javier Álvarez

**Affiliations:** 1Departamento de Bioquímica y Biología Molecular y Fisiología, Facultad de Medicina, Universidad de Valladolid, 47005 Valladolid, Spain; silvia.romero.sanz@uva.es (S.R.-S.); elena.caldero@uva.es (E.C.-E.); mariapilar.alvarez.illera@uva.es (P.Á.-I.); jaime.santo-domingo@uva.es (J.S.-D.); sergio.delafuente@uva.es (S.d.l.F.); paloma.garcia@uva.es (P.G.-C.); rosalba.fonteriz@uva.es (R.I.F.); mmontero@uva.es (M.M.); 2Unidad de Excelencia Instituto de Biomedicina y Genética Molecular (IBGM), Universidad de Valladolid y Consejo Superior de Investigaciones Científicas (CSIC), 47005 Valladolid, Spain

**Keywords:** *C. elegans*, rotenone, Parkinson’s disease, CGP37157, lifespan, mitochondria, Ca^2+^ signaling

## Abstract

We have previously shown that the compound CGP37157, a mitochondrial Na^+^/Ca^2+^ exchanger inhibitor, increases lifespan and improves muscle and mitochondrial structure during aging in wild-type *C. elegans* nematodes. We used here a rotenone model of Parkinson’s disease in *C. elegans* to test the ability of CGP37157 to rescue the alterations induced by the toxicant. Rotenone, a mitochondrial respiratory chain complex I inhibitor, reduced worm lifespan and muscle activity, measured as worm mobility, pharyngeal pumping, and defecation rate. It also increased ROS production, decreased mitochondrial membrane potential, and disorganized mitochondrial structure. Moreover, it induced degeneration of dopaminergic neurons and changes in behavior. We found that CGP37157 produced a partial or complete reversal of most of these alterations. These results are consistent with our previous proposal that Ca^2+^ homeostasis is important in the development of neurodegenerative diseases, and modulation of the Ca^2+^ signaling toolkit may be a novel target for their treatment.

## 1. Introduction

The increase in the number of elderly people in advanced societies is generating a steady rise in the number of cases of neurodegenerative diseases. Among them, the second in incidence is Parkinson’s disease (PD), a slowly progressive degenerative disorder that generates resting tremor, rigidity, slowness, bradykinesia, and instability. Anatomopathologically, it is characterized by the accumulation of Lewy bodies filled with α-synuclein aggregates in various parts of the nervous system, especially in the nigrostriatal system [1].

As for its origin, although there are some genes whose mutations are clearly associated with the disease, 90% of cases are sporadic and of unknown origin. In these cases, it is believed that its development depends on a combination of factors including age, individual genetic variability, and the influence of certain environmental agents [2]. Among the external agents that can act as triggers, we must mention a series of neurotoxins that generate symptoms analogous to those of PD in various animal models, and even in the human species when it has been exposed to these toxins [3]. Examples include MPTP, 6-OHDA, or the plant-derived insecticide rotenone [2,3,4,5,6]. Interestingly, all these compounds inhibit complex I of the mitochondrial respiratory chain [2,6,7] and as a consequence reduce ATP synthesis and increase ROS production at the mitochondrial level. The available evidence suggests in any case that the neurotoxic effects are due more to increased ROS production than to energy depletion [7,8]. It is also important to mention that these neurotoxins have facilitated the study of this disease because they have made it possible to generate animal models of PD that reproduce most of its characteristics [2,4,9]. In particular, the rotenone-induced PD model in *C. elegans* has been widely used [2,4,10,11,12].

Another factor that has been linked to the disease is an alteration of Ca^2+^ homeostasis, which would be associated with processes that promote neurodegeneration such as endoplasmic reticulum (ER) stress, impaired connections among ER, mitochondria, and lysosomes, excessive production of ROS, or altered autophagy phenomena [13]. Some of the proteins whose mutations are associated with PD, such as PINK1, parkin, or DJ-1, play an essential role in mitophagy [14]. Moreover, all these proteins and α-synuclein modulate ER–mitochondrial crosstalk and thus Ca^2+^ transfer between the ER and mitochondria, such that in various PD models, an increase in ER–mitochondrial Ca^2+^ transfer and an increase in mitochondrial Ca^2+^ level are observed [15,16]. It has been further described that α-synuclein aggregates activate the SERCA Ca^2+^ pump, which would contribute to increased ER–mitochondrial Ca^2+^ transfer as well [17].

We have recently described that some modulators of Ca^2+^ homeostasis such as SERCA inhibitors or the mitochondrial Na^+^/Ca^2+^ exchanger inhibitor CGP37157 are able to increase lifespan and health span in *C. elegans* nematodes [18,19,20,21]. We have also reported that SERCA inhibition reverses many of the alterations induced by rotenone in *C. elegans* worms [22], suggesting a role for ER Ca^2+^ or ER–mitochondrial Ca^2+^ transfer in rotenone toxicity. In view of the role of mitochondria and Ca^2+^ homeostasis in PD, we decided to investigate here whether the mitochondrial Na^+^/Ca^2+^ exchanger inhibitor CGP37157 could also be protective in a model of rotenone-induced PD in *C. elegans*. Our data show that indeed this inhibitor is able to fully or partially reverse many of the rotenone-induced alterations, suggesting that the exchanger plays a role in rotenone-induced alterations and could also be a target for disease treatment.

## 2. Results

We investigated the effect of adding the compound CGP37157 from day 1 of adult life on both nematode lifespan and several parameters representative of health span.

### 2.1. Effects on Lifespan

As we have previously reported [22], rotenone reduces the lifespan of the worms in two phases: an initial increase in mortality, which occurs immediately after the addition of the inhibitor, and a reduction in the lifespan of the rest of the surviving population after the initial phase (Figure 1). The addition of CGP37157 rescued the worms from both effects of rotenone. It reduced the initial mortality and restored the lifespan of the rest of the population to that of control worms (Figure 1). It should be noted that the concentration used, 25 µM, produced no effect in the controls, as reported previously [19], indicating that the worms treated with rotenone were more sensitive to the effect of the compound. Further details of all assays performed can be found in Appendix A.

### 2.2. Effects on ROS Production

As a consequence of the inhibition of complex I of the mitochondrial respiratory chain, rotenone causes an increase in ROS production [4,7,23], which appears to play an essential role in the development of the PD model induced by this toxicant [7,8]. It is therefore important to determine whether CGP37157 has any effect on mitochondrial ROS production. We measured them using MitoSox, and our data show that rotenone produced a large increase in ROS production that was completely reversed in the presence of CGP37157 (Figure 2A,B).

### 2.3. Effects on Mitochondrial Membrane Potential

As a parameter of mitochondrial health, mitochondrial membrane potential was also measured. It is also affected by rotenone, as would be expected from its action as a respiratory chain complex I inhibitor. Rotenone reduced mitochondrial membrane potential almost as much as the protonophore CCCP, and addition of CGP37157 fully restored mitochondrial membrane potential to control levels (Figure 2C,D).

### 2.4. Effects on Oxygen Consumption Rate

Given that the mitochondrial respiratory chain is the main site of ROS production and CGP37157 restored control ROS levels and mitochondrial membrane potential, we measured oxygen consumption by the worms in the different conditions to test if electron transport were also restored. Measurements of oxygen consumption rate (OCR) showed that rotenone induced a reduction in respiration rate under resting conditions, as would be expected for an inhibitor of complex I. However, this effect was not reverted in the presence of CGP37157. Figure 2E–G show the mean values of OCR obtained for basal, maximum, and non-mitochondrial respiration in each condition. CGP37157 reduced respiration even further in the presence of rotenone, although it produced no effects in the absence of rotenone. Therefore, the improvements in other parameters of health were not due to an overriding of the blockage of the respiratory chain.

### 2.5. Effects on Mitochondrial Structure

The mitochondrial structure of body wall muscle cells undergoes a progressive degeneration with age. These changes can be followed by using a *C. elegans* strain expressing GFP in body wall muscle mitochondria. In young adult nematodes, mitochondria show a clean and orderly parallel appearance. Then, as they age, aggregates and discontinuities begin to form, resulting in a progressively disordered appearance. We have reported before that the effects of CGP37157 on the longevity of wild-type worms requires the integrity of the mitochondrial respiratory chain [19,21], and we have also seen that this compound delays the disorganization of the mitochondrial structure in wild-type worms [21]. For these reasons, we decided to study the effects of CGP37157 on the mitochondrial structure of the PD model.

In wild-type nematodes, these changes start to be seen mainly from day 8 onwards. However, in the presence of rotenone, they appear early and are already evident on day 5 (Figure 3). When the worms are incubated in the presence of both rotenone and CGP37157, the structure apparently recovers a more ordered and parallel structure. These changes can be further studied by using image analysis programs to calculate morphological parameters such as mitochondrial number per cell, median branch length, number of branches per mitochondria, or form factor [22]. Figure 3 shows that rotenone reduced the number of mitochondria per cell and the branch length and increased the number of branches per mitochondria and the form factor, as we have previously reported [22]. After treatment with CGP37157, most of the parameters were unchanged, but there was a significant recovery in the mean branch length.

### 2.6. Effects on Dopaminergic Neurons

Degeneration of brain dopaminergic neurons plays a key role in the development of Parkinson’s disease. *C. elegans* worms have four bilaterally symmetric pairs of dopaminergic neurons, three of them within the head. These neurons possess ciliated dendrites that contact the cuticle and transmit mechanosensory information. We used the strain BZ555, inserting the GFP gene specifically tagged to the *C. elegans* dopamine transporter gene (*dat-1*), which is expressed only in the dopaminergic neurons [24]. This strain has intense GFP expression in all dopaminergic neurons and has been used to detect degeneration of these neurons from the reduction in the intensity of GFP fluorescence [25,26]. Our data show that treatment with rotenone significantly reduced neuronal fluorescence, an indication of neuronal degeneration, and treatment with CGP35157 led to a partial recovery of the fluorescence (Figure 4A). A typical image of neuronal fluorescence is shown in Appendix A.

### 2.7. Effects on Associative Learning

It has been reported that *C. elegans* worms placed in a plate containing ethanol in two opposite quadrants try to avoid the ethanol-containing quadrants [27]. However, after 4 h preexposure in plates containing ethanol, they instead show preference for the ethanol-containing quadrants. This behavior has been suggested as a model of complex associative learning. Moreover, dopamine synthesis was shown to be required for this behavior [27], suggesting that it could be altered in our PD model. Figure 4B shows that ethanol preexposure increased ethanol preference in control worms. This behavior disappeared in the presence of rotenone (Figure 4C) and was restored by CGP37157 (Figure 4D).

### 2.8. Effects on Muscle Activity: Mobility, Pharynx Pumping, and Defecation

Worm mobility was assessed by tracking the movement of worms in short videos and calculating the speed of movement of each worm, either the maximum speed, the average speed, or the speed measured in body lengths per second (BLPS), to account for differences in worm size. Figure 5A–C show that rotenone reduced the speed measured by each method, and the addition of CGP37157 produced a significant increase in speed at day 8 of adult life.

In addition to studying the motility of worms in tracking studies, other measures of muscle activity in *C. elegans* are the rate of pharyngeal pumping and the rate of defecation (Figure 5D,E). Pharyngeal pumping was largely (over 90%) reduced by rotenone. The addition of CGP37157 partially restored it, although still far from the rate observed in control worms. Defecation also has a characteristic rhythm in *C. elegans* that depends on the activity of several stages of muscle wall contraction [28]. Our data show that rotenone caused an almost 90% decrease in defecation rate and CGP37157 increased it by about threefold compared to the value in the presence of rotenone, but again still far from the values in the controls (Figure 5E).

### 2.9. Effects on Electropharyngeogram

An electropharyngeogram (EPG) measures the rhythmic electrical activity of the pharyngeal muscles responsible for their rhythmic contraction to ingest food, i.e., pharyngeal pumping. It is therefore a different and more sensitive method of monitoring pharyngeal pumping, allowing direct measurement of parameters such as pump duration and intensity of the depolarization and repolarization waves. We performed experiments on control worms and worms treated with rotenone or rotenone + CGP37157, and the mean parameters obtained are shown in Figure 6. Consistently with the pumping rate measurements shown in Figure 5D, the mean pumping frequency was also reduced by rotenone and the effect of rotenone was partially reversed in the presence of CGP37157. In addition, both the mean pump duration, i.e., the interval between depolarization and repolarization, and the R/E ratio, i.e., the ratio of repolarization and depolarization waves in each pump, were increased by rotenone and partially restored by CGP37157.

### 2.10. Effects on Size and Fertility

Worms treated with rotenone showed a significant reduction in size, with a reduction in worm area of almost 50%. In this case, however, the addition of CGP37157 did not lead to any changes in size (Appendix A). Similarly, fertility was greatly reduced after rotenone treatment. Here again, the addition of CGP37157 did not produce a significant increase in fertility in rotenone-treated worms (Appendix A).

## 3. Discussion

Benzothiazepine CGP37157 is considered a selective inhibitor of the mitochondrial Na^+^/Ca^2+^ exchanger (mNCX), the main pathway for calcium extrusion from the mitochondria. However, it has been described that it can also act on other calcium transport systems. Among these, it has been described to inhibit L-type and CALHM1 voltage-dependent Ca^2+^ channels [29,30] and plasma membrane Na^+^/Ca^2+^ exchangers [31], in all cases at concentrations similar to those required to inhibit mNCX. Therefore, the beneficial effects of this compound may be due to a mixture of effects on different systems, although all of them related to Ca^2+^ homeostasis.

It has been described that CGP37157 can act as a neuronal protector in several models of neuronal toxicity. Specifically, CGP37157 rescued neuronal death induced by veratridine, glutamate, or ischemia/reperfusion in rat chromaffin cells and hippocampal slices [32,33,34]. It also protected cultures of SH-SY5Y human neuroblastoma cells subjected to high-potassium stimulation [35] and primary cultures of NMDA-stimulated rat cortical neurons [36]. CGP37157 has also been recently shown to protect striatal neurons from α-synuclein plus rotenone-induced toxicity [37]. In addition, we have recently described that CGP37157 treatment extends the lifespan of wild-type *C. elegans* worms and also improves a number of health parameters, such as mobility or the regular structure of sarcomeres and mitochondria in body wall muscle [19,21]. These effects disappeared in mutants of respiratory chain complex I and in mTOR pathway mutants, suggesting that integrity of the mitochondrial respiratory chain and mTOR pathway were important for the effects.

For all these reasons, we thought that CGP37157 could also act as a protector in a model of neurodegenerative disease. We chose the model of Parkinson’s disease induced by rotenone in *C. elegans*. Rotenone generates in various animal species a series of alterations that mimic those found in PD [2,4,9]. In worms, rotenone also generates a series of structural, motor, and behavioral alterations that are consistent with alterations that appear in PD [2,4,10,11,12,38]. In this work, we found that CGP37157 totally or partially reverses many of the alterations induced by rotenone, so it could be a good candidate in the study of possible new treatments for this disease.

First, rotenone reduces the half-life of wild-type N2 worms, and CGP37157 restored it to the same values as in controls. CGP37157 was also very effective in blocking the increase in ROS induced by rotenone. In fact, the increase in ROS was completely reversed to control values. This is important, because the increase in ROS is considered to be the main mechanism responsible for the effects of rotenone, rather than energy depletion following the inhibition of complex I of the respiratory chain [7,8]. This blockade of ROS increase was not due to a reactivation of the respiratory chain, since CGP37157 was not able to reverse the decrease in oxygen consumption rate (OCR) induced by rotenone. However, in spite of this, CGP37157 fully restored mitochondrial membrane potential. Altogether, this suggests that CGP37157 promotes a better mitochondrial coupling efficiency and a smaller electron leak and ROS production. Consistently, CGP37157 also produced beneficial effects in mitochondrial structure, which is significantly altered in the presence of rotenone.

Rotenone also induces neurological alterations, in particular degeneration of dopaminergic neurons, a defining feature of PD. This effect can be directly monitored by using a *C. elegans* strain expressing GFP in the dopaminergic neurons, and our data show that CGP37157 reduced the degeneration of dopaminergic neurons induced by rotenone. Moreover, we have also studied the effects of rotenone and CGP37157 in a model of ethanol-induced complex worm behavior, which is dependent on dopamine synthesis [27]. Consistently, we found that CGP37157 also reversed the changes induced by rotenone in worm behavior in this model.

Rotenone also considerably reduces worm mobility in tracking studies, and CGP partially rescued this effect by enhancing worm mobility again, although without reaching the level observed in controls. The same was observed in the rates of pharynx pumping and defecation, which decreased by about 90% in the presence of rotenone, and CGP37157 also managed to increase them, although at levels still far from those of the controls.

EPG measurements also showed a significant decrease in frequency induced by rotenone (Figure 6A), but not as large as the decrease in pumping rate (Figure 5D), suggesting that in some cases, the electrical activity we measured may not be able to trigger contraction. The decrease in frequency was accompanied by an increase in both pump duration and R/E ratio. The increase in the R/E ratio means that rotenone reduces the depolarization wave, i.e., it reduces the electrical excitation of the pharyngeal muscle. On the other hand, the increase in pump duration means that repolarization is delayed. Electrical excitation is triggered by MC cholinergic neurons, and repolarization is partly mediated by M3 glutamatergic neurons. Our results suggest that rotenone appears to affect both types of neurons and CGP37157 partially reverses this toxic effect. The decrease in electrical excitation induced by rotenone may explain why in some cases the electrical pumps do not trigger physical contractions.

Finally, fertility was largely reduced in the presence of rotenone and CGP37157 increased it again significantly, but still far from the values of the controls. CGP37157 was unable to recover the size of the worms, which was nearly halved by rotenone.

Overall, we found that CGP37157 ameliorates many of the alterations induced by rotenone in many different health parameters in *C. elegans*, including lifespan, muscle, and neuronal and mitochondrial function, all characteristics of this Parkinson’s disease model. Of course, this model has some limitations. Although *C. elegans* shares more than 70% of its genome with us, it is very different. It lacks many mammalian organs, such as the circulatory system, and has a very simple nervous system, which cannot be compared with the complexity of the human nervous system. Nevertheless, in many cases, data from *C. elegans* have predicted phenomena that were later found in other models of neurodegeneration [39]. In any case, these results should be validated in mammals as a starting point before moving on to preclinical and clinical studies, and this is one of our main aims.

Like all models, it has advantages and limitations, and the data obtained must go through the filter of validation in other models before being applied to humans. However, the utility of the *C. elegans* model in terms of carrying out studies quickly and easily with large numbers of individuals and at low cost is a major advantage for these studies.

Regarding the mechanism of the effect, we have previously found that partial inhibition of the SERCA pump is also able to rescue alterations in this rotenone model of PD [22]. Our data here show that another modulator of Ca^2+^ homeostasis produces the same effects. Although CGP37157 may not be completely specific, all the known targets of this molecule belong to the Ca^2+^ homeostasis machinery, either the mitochondrial Na^+^/Ca^2+^ exchanger or several Ca^2+^ channels in the plasma membrane. The effects of SERCA inhibition may be mediated by a reduction in the ER Ca^2+^ concentration and ER–mitochondria Ca^2+^ transfer. The effects of CGP37157 may be more complex, as it would combine inhibition of both Ca^2+^ entry into the cytosol and Ca^2+^ efflux from the mitochondria, resulting in a decrease in both mitochondrial Ca^2+^ influx and efflux. In both cases, the interplay between mitochondrial, cytosolic, and ER Ca^2+^ levels may interact with pathways such as mTOR to promote longevity or health. Therefore, the Ca^2+^ homeostasis toolkit appears to be a promising target to explore in age-related diseases.

## 4. Materials and Methods

### 4.1. C. elegans Strains and Maintenance

*C. elegans* nematodes were maintained and handled as previously described [40]. Strains were maintained at 20 °C. Strains SJ4103 (zcls14(myo-3::GFP(mit)), a N2-derived strain that expresses mitochondrial GFP in body wall muscle cells) and BZ555 (dat-1p::GFP), a N2 derived strain that expresses the fluorescent protein GFP in dopamine neuronal soma and processes, were obtained from the Caenorhabditis Genetics Center (CGC, University of Minnesota, MN, USA).

### 4.2. Administration of Rotenone and CGP37157

When required, CGP37157 was dissolved in NGM agar at 50 µM and rotenone was added over the bacterial lawn at 14 µM. Young adult nematodes (day 1) were then transferred to the plates for the different assays.

### 4.3. C. elegans Lifespan Assays

Lifespan assays were carried out as previously described [21]. Briefly, *E. coli* (OP50)-seeded NGM 35 mm plates were prepared containing 15 μM fluorodeoxyuridine (FUdR) to avoid progeny and CGP37157 or rotenone when required. Then, around 100 synchronized young adults (day 1) per condition were transferred to the plates (10 worms/plate). Control and drug-containing assays were always carried out in parallel at 20 °C in a temperature-controlled incubator. Plates were scored for dead worms every day. Other details were as described previously [21].

### 4.4. Mitochondrial ROS

Mitochondrial ROS were measured in 5-day-old N2 worms grown in the presence or in the absence of rotenone and/or CGP37157. A total of 20–30 worms were collected, washed with M9 medium, and incubated with shaking for 4.5 h in the dark with 500 µL of M9 medium containing 10 µM MitoSOX. Then, after washing with M9 medium, 7–10 worms were placed on an agar pad with a drop of 10 mM tetramisole for fluorescence measurement with a Leica TCS SP5 confocal microscope (Leica, Madrid, Spain) (510 nm/580 nm excitation/emission wavelengths). Fluorescence intensity was proportional to the level of mitochondrial ROS.

### 4.5. Mitochondrial Membrane Potential

Mitochondrial membrane potential was determined in 5-day-old N2 worms grown in the presence or in the absence of rotenone and/or CGP37157. A total of 20–30 worms were collected, washed with M9 medium, and incubated with shaking for 3.5 h in the dark with 500 µL of 0.1 µM TMRE in M9 medium. A fully depolarized control was created by adding 100 µM of the protonophore CCCP. Then, after washing with M9 medium, 7–10 worms were placed on an agar pad with a drop of 10 mM tetramisole for fluorescence imaging in a Nikon ECLIPSE Ni-E microscope (Nikon, Barcelona, Spain) equipped with a TRITC filter and a DS-Ri2 camera. Fluorescence intensity was proportional to the mitochondrial membrane potential.

### 4.6. Measurement of Oxygen Consumption Rate

Nematode oxygen consumption rate (OCR) was measured with the Seahorse Xfe24 extracellular flux analyzer (Agilent Technologies, Inc., Agilent, Madrid, Spain), as described before (Luz et al., 2015) [41], using 24-well Seahorse XF24 V7 PS cell culture microplates (Agilent, Madrid, Spain). Thirty minutes before the experiment, day 5 synchronized adult worm populations were washed twice with M9 medium and put into a Seahorse tissue culture plate at a density of 10–12 worms per well. Respiration rates were measured every 5 min using the following protocol: 2 min mixing, 30 s waiting, and 2 min measuring. All experiments were performed at 25 °C.

### 4.7. Mitochondrial Structure

Mitochondrial structure was studied by confocal imaging in 5-day-old SJ4103 grown in the presence or in the absence of rotenone and/or CGP37157. They were transferred to a 2% agarose pad containing a drop of 50 mM sodium azide, covered with a coverslip, and imaged on a Leica TCS SP5 confocal microscope (excitation 488 nm, emission 500–554 nm). Images were then processed and analyzed using ImageJ software v. 1.54k and the plugin Mitochondria Analyzer [42].

### 4.8. Fluorescence from Dopaminergic Neurons

The integrity of dopaminergic neurons was measured indirectly by assessing the fluorescence of dopaminergic neurons in the BZ555 strain. For every experiment, 10 synchronized day 5 adult-stage BZ555 worms were transferred onto agarose pads. Z-stacks of CEP and ADE dopaminergic neurons were acquired using epifluorescence microscopy (Eclipse Ni-E (focusing nosepiece system)) with an attached Nikon DS-Ri2 camera, 20× objective, and a FITC filter cube (475 nm/509 nm excitation/emission wavelengths), all from Nikon, Barcelona, Spain. Around 20 Z-stack images of 1 µm were projected using the v. 1.54k ImageJ tool Z-projection to generate complete dopaminergic neuron images. To quantify the mean fluorescence intensity of the somas, they were chosen as ROIs using a threshold that was applied to all images.

### 4.9. Associative Learning

Associative learning was assessed as described before [27]. Briefly, 5-day-old N2 worms grown in the presence or in the absence of rotenone and/or CGP37157 were placed for 4 h in plates either with or without 300 mM ethanol. After washing with M9 buffer, worms were placed in the center of new plates divided into four quadrants, with two of them diagonally opposite containing 10 μL of 300 mM ethanol. After 30 min, the preference index (PI) was calculated as PI = (worms in ethanol quadrants − worms in control quadrants)/total worms.

### 4.10. Tracking and Mobility

Mobility assays were performed in 5- to 8-day-old N2 worms grown in the presence or in the absence of rotenone and/or CGP37157. For each condition, one-minute videos were taken and analyzed offline using the v. 1.54k ImageJ plugin wrMTrck, as described previously [43]. This analysis allowed us to obtain the average speed, the maximum speed, the speed measure as body lengths per second (BLPS), and the area of the worms.

### 4.11. Pharynx Pumping

Pharynx pumping was measured in 3-day-old N2 worms grown in the presence or in the absence of rotenone and/or CGP37157 using a stereomicroscope (Leica S9d, Leica, Madrid, Spain) at 40× magnification. Ten worms from each condition were measured, and each worm was counted for 20 s every minute for 10 min.

### 4.12. Defecation

Defecation was measured in 3-day-old N2 worms grown in the presence or in the absence of rotenone and/or CGP37157. For at least 10 worms per condition, 30 min videos were recorded and used to count the number of defecations.

### 4.13. Electropharyngeogram

This technique allows the recording of electrical activity during pharyngeal pumping, as previously described [44]. Briefly, a Nemametrix screen chip system was placed on an inverted Zeiss Axiovert 200 microscope (Zeiss, Madrid, Spain) equipped with an LD A-Plan 20× objective. Worms were washed first with 1.5 mL of 0.2 µm-filtered M9 buffer + 0.1% Tween, then 4× with 0.2 µm-filtered M9 buffer, once in M9 buffer containing 2.3 mM serotonin, and finally suspended in 1 mL of M9 buffer containing 2.3 mM serotonin. All the experiments were made between 15 and 120 min after serotonin exposure. A fresh screen chip containing M9 buffer with serotonin was introduced in the system and loaded with worms to start recording. Then, the electrical activity of each worm was recorded for 3 min, and more than 30 worms were assayed for each condition.

### 4.14. Fertility

Fertility was measured in N2 worms grown without FUdR and in the presence or in the absence of rotenone and/or CGP37157. They were placed (1 worm/well) in 24-well plates and to new wells every day for 4 days. After that, eggs and larvae in each well were counted.

### 4.15. Materials

Rotenone and the protonophore CCCP (carbonyl cyanide 3-chlorophenylhydrazone) were obtained from Sigma, Madrid, Spain. CGP37157 was from Tocris Bioscience (Bristol, UK). TMRE (tetramethylrhodamine, ethyl ester) and MitoSOX red were obtained from Molecular Probes (Thermo Fisher Scientific, Madrid, Spain). FuDR was acquired from Alfa Aesar, Karlsruhe, Germany. Other reagents were from Sigma, Madrid, Spain or Merck, Darmstadt, Germany.

### 4.16. Statistical Analysis

In the lifespan assays, statistical analysis was performed with SPSS software (v. 22) using the Kaplan–Meier estimator and the log-rank routine for significance of each assay. Three assays per condition were conducted, and the half-lives were compared using one-way ANOVA and Tukey’s test. The significance of each assay is shown in Appendix A. In the rest of the studies, data are shown as means ± s.e.m., and significance was calculated using one-way ANOVA (F and P parameters are shown in Appendix A) and post hoc comparisons with Tukey’s test, except for some assays, where the *t*-test was used, as mentioned in the figure legends.

## Figures and Tables

**Figure 1 ijms-26-03371-f001:**
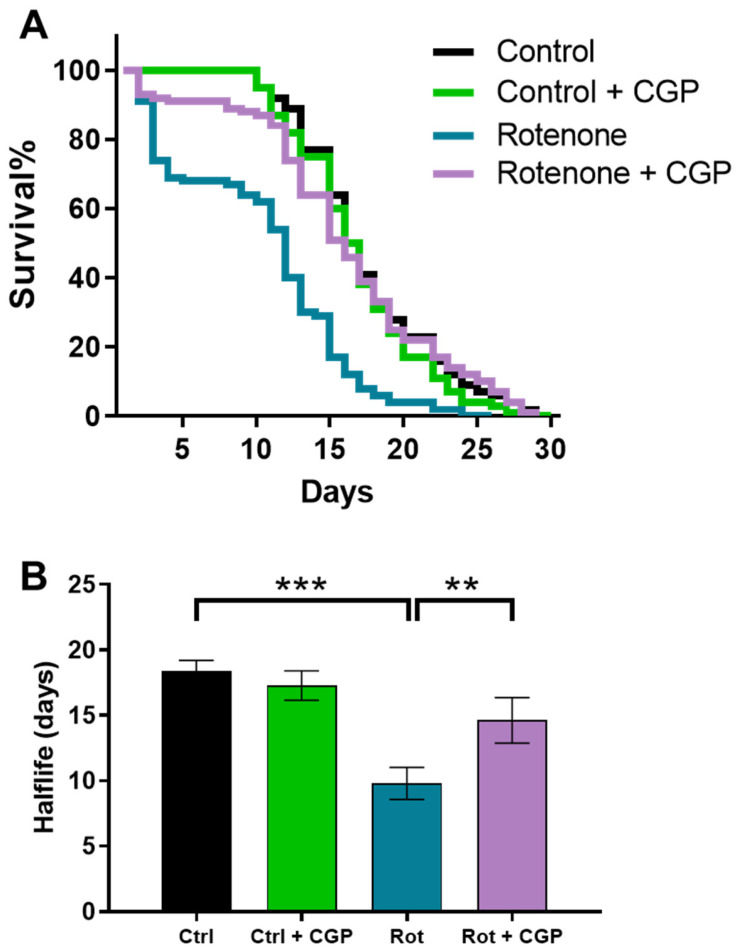
Reversal by CGP37157 of the effect of rotenone on *C. elegans* lifespan. (**A**) Typical survival curves obtained in the N2 strain in control worms, worms treated with rotenone, and worms treated with rotenone + CGP37157. (**B**) Mean half-life obtained in 3 similar assays of each kind (see Appendix A for more data on the assays). Data in (**B**) are means ± s.e.m. Statistics derived by ANOVA and means comparisons by Tukey’s test (** *p* < 0.01; *** *p* < 0.005). Appendix A shows F and P parameters for all the ANOVAs.

**Figure 2 ijms-26-03371-f002:**
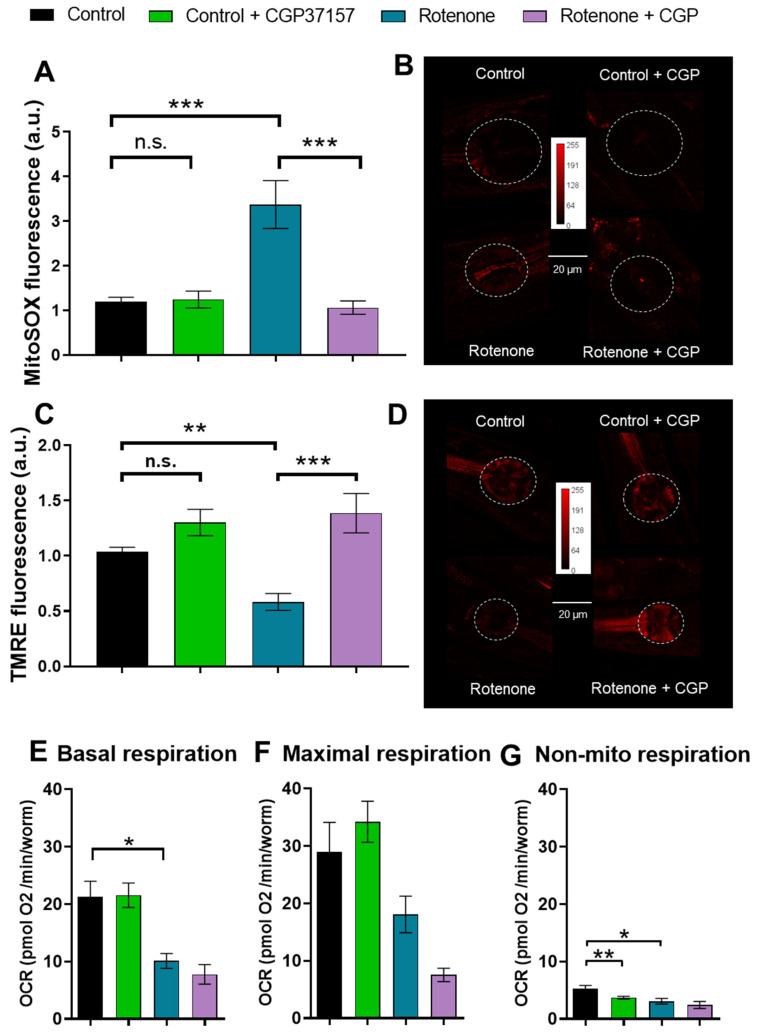
Reversal by CGP37157 of the effect of rotenone on mitochondrial ROS production and mitochondrial membrane potential, but not on oxygen consumption rate (OCR). (**A**,**B**) Mitochondrial ROS measurements with mitoSOX. (**A**) Means of 3 different experiments in each condition. (**C**,**D**) Mitochondrial membrane potential measurements with TMRE. The effect of complete depolarization with the protonophore CCCP is shown in Appendix A. (**C**) Means of 3 different experiments in each condition. Representative images of mitoSOX or TMRE fluorescence in each condition are shown in (**B**,**D**). (**E**–**G**) OCR measurements, all obtained in control worms, worms treated with rotenone, or worms treated with rotenone + CGP37157. In OCR measurements, the protonophore FCCP was added to obtain the maximum rate of respiration and azide to obtain non-mitochondrial respiration. For OCR measurements, 3 different experiments in each condition were conducted on 3 different days. Data are means ± s.e.m. Significance was measured using Tukey’s test after ANOVA. * *p* < 0.05; ** *p* < 0.01; *** *p* < 0.005; n.s., not significant.

**Figure 3 ijms-26-03371-f003:**
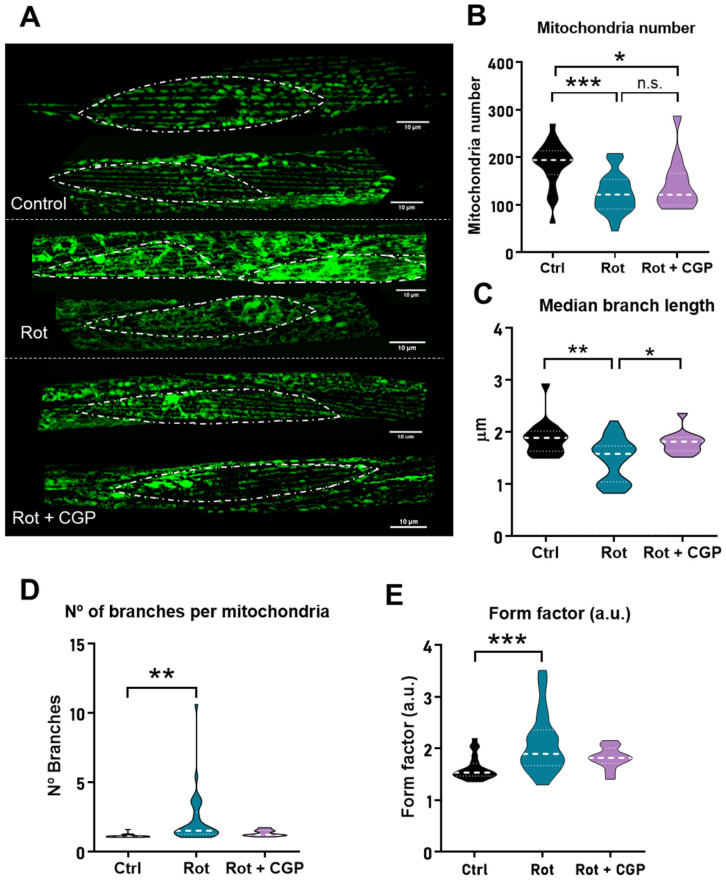
Effect of rotenone and CGP37157 on mitochondrial structure. Experiments were conducted on the SJ4103 *C. elegans* strain expressing mitochondrial GFP in body wall muscle. (**A**) A series of representative confocal images of mitochondrial fluorescence are shown in the three different conditions: control worms (n = 21), worms treated with rotenone (n = 30), and worms treated with rotenone + CGP37157 (n = 13). Bar plots show the morphological analysis of the images obtained in each condition: the mitochondrial number per cell (**B**), the median branch length (**C**), the number of branches per mitochondria (**D**) and the form factor (**E**). Statistics derived by ANOVA and means comparisons by Tukey’s test. * *p* < 0.05; ** *p* < 0.01; *** *p* < 0.005; n.s., not significant.

**Figure 4 ijms-26-03371-f004:**
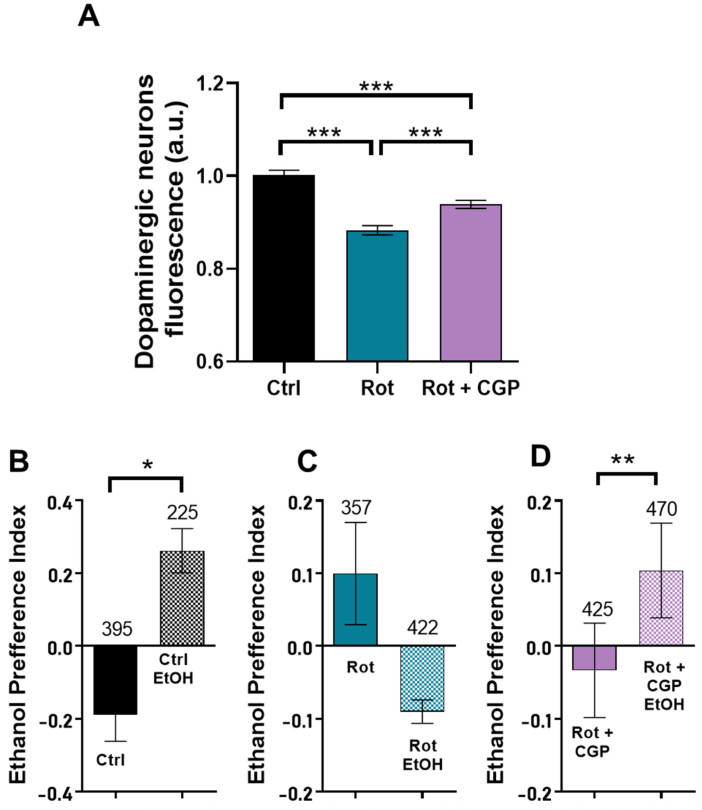
Reversal by CGP37157 of the effect of rotenone on dopaminergic neurons and ethanol preference. (**A**) Changes in the fluorescence of the dopaminergic neurons of the BZ555 strain induced by rotenone or rotenone + CGP37157. Data are means of 26–30 different worms in each condition. (**B**–**D**) Effect of ethanol preexposure on ethanol preference in the three different conditions: control N2 worms (**B**), N2 worms treated with rotenone (**C**), and N2 worms treated with rotenone + CGP37157 (**D**). Data are means ± s.e.m. of 4–7 different experiments of each kind. Significance was measured using Tukey’s test after ANOVA (**A**) and *t*-tests (**B**–**D**). * *p* < 0.05; ** *p* < 0.01; *** *p* < 0.005. The number of worms tested is indicated above each bar.

**Figure 5 ijms-26-03371-f005:**
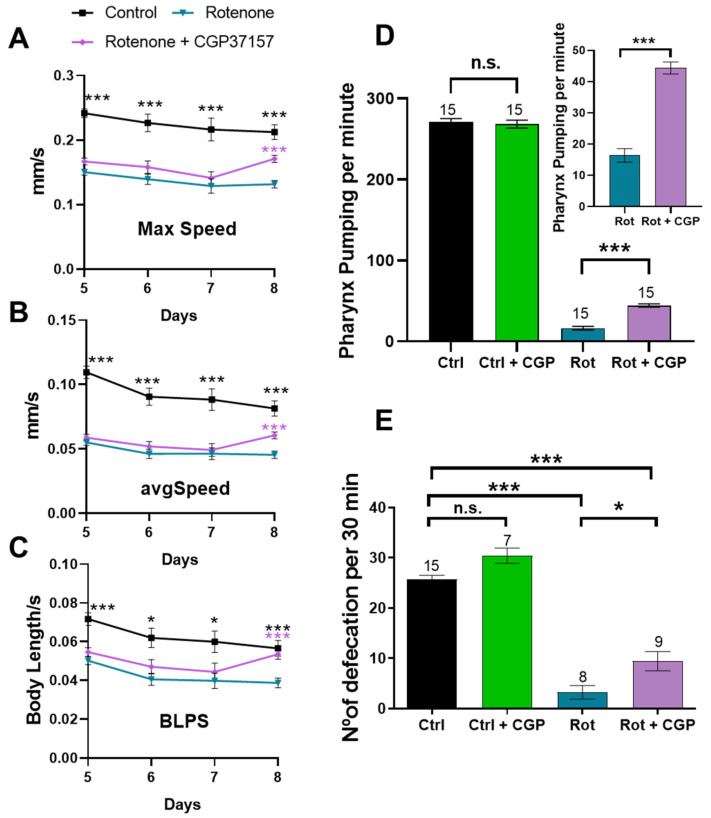
Reversal by CGP37157 of the effect of rotenone on *C. elegans* mobility, pharyngeal pumping, and defecation. Maximum speed (**A**), average speed (**B**), and speed measured as body length per second (BLPS; (**C**)). Data were obtained on days 5–8 of adult life in 3 independent experiments that included measurements of 50 control worms and 120 worms treated with either rotenone or rotenone + CGP37157. Statistical significance is shown for control vs. rotenone (black *) and rotenone vs. rotenone + CGP37157 (violet *). (**D**,**E**) Rate of pumping and rate of defecation, respectively, in either control worms or worms treated with rotenone, in both cases in the absence or in the presence of CGP37157. Data were obtained by counting the pumping rate or the defecation rate in day 3 worms. The number of worms studied in each case is indicated on top of each bar. Data are means ± s.e.m. Statistics were derived by ANOVA and means comparisons by Tukey’s test. * *p* < 0.05; *** *p* < 0.005; n.s., not significant.

**Figure 6 ijms-26-03371-f006:**
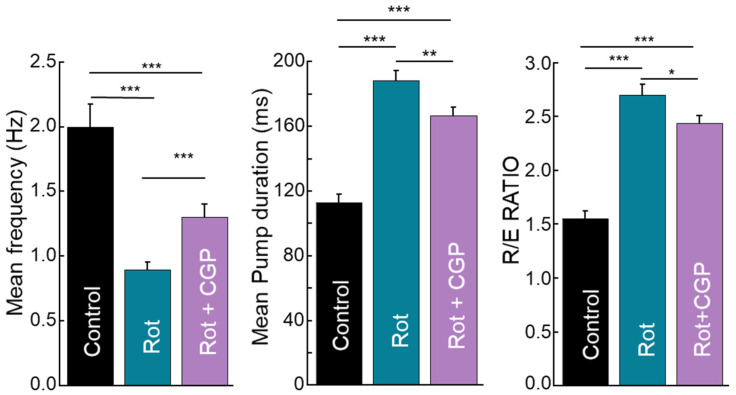
Reversal by CGP37157 of the effect of rotenone on *C. elegans* electropharyngeogram. The figure shows mean data obtained from a series of electropharyngeograms performed on control worms (36 worms analyzed), worms treated with rotenone (64 worms analyzed) or with rotenone and CGP37157 (51 worms analyzed). Mean frequency in hertz (**A**), mean pump duration in milliseconds (distance from depolarization to repolarization) (**B**), and R/E ratio (ratio of the amplitude of the repolarization to that of the depolarization) (**C**). Statistics were derived by ANOVA and means comparisons by Tukey’s test. * *p* < 0.05; ** *p* < 0.01; *** *p* < 0.001.

## Data Availability

The raw data supporting the conclusions of this manuscript will be made available by the authors without undue reservation to any qualified researcher.

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
