# Peer review of "Rescue of a Rotenone Model of Parkinson’s Disease in C. elegans by the Mitochondrial Na+/Ca2+ Exchanger Inhibitor CGP37157"

_ijms, 2025, doi:10.3390/ijms26073371_

Round 1
Reviewer 1 Report
Comments and Suggestions for Authors
In the manuscript “Rescue of a Rotenone model of Parkinson’s Disease in C. elegans by the mitochondrial Na+/Ca2+ exchange inhibitor CGP37157” by Romero-Sanz et al., the authors evaluated the efficacy of the compound CGP37157 in ameliorating multiple physiological and behavioral parameters in a C. elegans model of Parkinson’s Disease. (PD). The authors measured multiple parameters including lifespan, muscle activity, mobility, pharyngeal pumping, associative learning, oxygen consumption rate etc. with or without CGP37157 and the results revealed the neuroprotective effect of CGP37157. Overall, majority of the experiments were designed, executed, interpreted and described properly, although there are few caveats that need to be addressed. Specific comments are summarized below.
- While some experiments have the ‘Control + CGP37157’ sample, there are experiments where this sample is missing, and the comparisons were made between control, rotenone and rotenone + CGP37157 sets. The reason is not clear. Control + CGP37157 sample is important and should be included in all the experiments unless there is a strong reason not to do so.
- Section 2.5 – ‘Effects on mitochondrial structure’ is a clear diversion. This is not relevant in this context and doesn’t add any value to overall conclusion. This sub-section should be removed. Similarly, the rationale behind doing the experiments described in section 2.8 and 2.9 is not very clear. This should either be explicitly mentioned or removed from the manuscript.
- Figure 2C. The CCCP data stands odd in the figure panel and should be moved as supplementary figure.
- Please include a representative image for Fig. 4A in the supplement.
- 5D. I am not convinced that the change between ‘rotenone’ and ‘rotenone + CGP’ is an appreciable change. I understand it is statistically significant but still far below the control.
- There are too many references and not every single reference is important/needed. The list of references should be shortened.
Author Response
Answer to Reviewer 1.
Please see enclosed File

Reviewer 2 Report
Comments and Suggestions for Authors
Dear authors and editor(s),
After reading this manuscript, my impressions were mostly positive, but I would like to point out a fundamental flaw in this report.
I have no idea which statistical tests were used. The Materials and Methods section lacks a description of the data analysis and statistics. In the Results section, some p-values are given, but it is not clear from the entire text, including the figure legends, which statistical tests were performed (except for two hints in Figures 2 and 4, where it says "T-test"). On this basis, the manuscript does not meet the minimum criteria for a research article, so I have no choice but to recommend rejection at this point.
I see no point in adding my other comments now. However, if the problem with the statistics is successfully resolved, I will include my other (mostly minor) comments in the next report.
Comments on the Quality of English LanguageA minor revision of the English language by a native English speaker is recommended.
Author Response
Answer to Reviewer 2
Please see enclosed File.

Reviewer 3 Report
Comments and Suggestions for Authors
This paper shows some interesting effects of CGP37157 on the health of C. elegans, particularly in models of Parkinson's disease. Although these results are promising, there are several points that need further elaboration to make the manuscript stronger and more impactful.
1) Molecular Mechanisms: Even though positive effects are observed, the mechanisms in play are not adequately investigated. A detailed visualization of the major pathways with specific focus on the targets of CGP37157 and its related signaling pathway, is needed. This will entail further exploring the putative targets of CGP37157 along with its related pathways and how these pathways may mediate the observed improvements in C. elegans health. In general, I would like to see experimental evidence supporting the mechanisms proposed.
2) Limitations of the Model: C. elegans is used exclusively to build upon one another. The findings should be compared with those of different models of parkinsonism (such as murine and cellular models), evaluated for similarities and differences, and for their potential for translation. Where such data exist, it is must acknowledge clearly and discuss the limitations of the reliance solely on C. elegans.
3) Long-lasting Effects: The study focuses on the short term. To evaluate the therapeutic potential of CGP37157, it is essential to bring are long-term investigation of CGP37157 on aging and neuronal health in C. elegans (lifespan, motility, and neuroprotection). It is crucial to come to an in-depth assessment of whether findings from C. elegans could inform treatment strategies in patients with Parkinson's disease. This should include potential challenges as well as opportunities for advancing CGP37157 or other related compounds as therapeutic agents are discussed. In that case, it may help to consider available preclinical and clinical studies, while bearing in mind the difficulties of generalizing findings from nematodes to humans. Resolving these issues will significantly strengthen the manuscript and will render a more significant and meaningful contribution to the field.
Author Response
Answer to Reviewer 3
Please see enclosed File

Round 2
Reviewer 1 Report
Comments and Suggestions for Authors
The authors have made necessary corrections in the revised manuscript. I recommend that the revised manuscript can be accepted for publication.
Author Response
Thank you very much for the comments.
Reviewer 2 Report
Comments and Suggestions for Authors
Dear authors and editors,
- The revised manuscript provided to me has no trackable changes, so it is difficult to effectively see the differences from the first version. For example, the number of cited references dropped from 51 to 44, without any explanation or labelling.
- In the Material and Methods section of the revised manuscript, the authors have added a paragraph on statistical analysis (lines 472-280). Assuming that the assumptions of ANOVA (independence of errors, random sampling, normality of distribution and homogeneity of variance) are met, three important questions arise here. First, if the authors have opted for ANOVA, the results of ANOVA (at least F and p values) should be presented. Nowhere in this manuscript are they presented. Only the results of the post-hoc tests are presented. Second, which ANOVA was used? A one-way ANOVA for all results? Or was a two-way ANOVA and a RM one-way ANOVA also used? Thirdly, if the ANOVA shows a significant difference between the group means, a suitable post-hoc test for multiple comparisons should be performed. The Tukey’s test is suitable here, but not the Student’s t-test. Multiple comparisons using t-tests increase the rate of type 1 (false positive) error, i.e. the rejection of the null hypothesis even though it is actually true.
- Why do the authors interpret the results contrary to the statistics? Lines 206-208: “Figure 5A‐C shows that rotenone reduced the speed measured by each method, and the addition of CGP37157 increased the speed at each day of life measured, although the effect was only significant at day 8 of adult life.” The authors claim that speed is increased before day 8, although there is no increase, i.e. no significant difference.
- Experimental design. In some experiments there are four groups (Ctrl, Ctrl+CPG, Rot, Rot+CPG), while in others there are three groups (Ctrl, Rot, Rot+CPG). Why is this so?
- All tables and figures should be self-sufficient, including those in the supplementary material. They should contain information about the statistical tests whose results are presented. This does not apply to the table and figures in the supplementary material of this manuscript.
- Supplementary Fig. S1. Which statistical test was performed here?
- Lines 385-386: “incubated with shaking for 4,5h in the dark with 500μl of 10 μM MitoSOX in M9 medium.” The volume and concentration of the MitoSOX solution are meaningless if we do not know the volume of the M9 medium. What was the concentration of MitoSOX? (Also, the authors use “.” and “,” interchangeably as decimal separators throughout the manuscript, as here in “4,5h”).
- A few comments on Figure 2, for example: While diagrams A and C show four groups, only three groups can be seen in diagrams B and D. Why is that? The CCCP group was removed from diagram C and shown in the supplementary material, but the label “CCCP” remained in the center of Figure 2. Besides that “Significance was obtained using ANOVA test” is not the best choice of words by itself, furthermore significant differences presented by asterisks were “obtained” using the Tukey’s test, not ANOVA.
- Why does the section on results consist of a mixture of introduction, materials and methods and results, with multiple references cited?
Comments on the Quality of English Language I would recommend a minor revision by a native English speaker.
Reviewer 3 Report
Comments and Suggestions for Authors
Thank you for your thorough work on the manuscript revision. All issues have been properly addressed, and I recommend acceptance in its current form.
Author Response
Thank you very much for the comments.
Round 3
Reviewer 2 Report
Comments and Suggestions for Authors
Please see the attachment.

A small revision of the English language is needed. For example, instead of "advanced societies" in line 30, I think "developed countries" would be a better choice of words.
Author Response
Please find the answer in the enclosed file.
